# Heat-Stress-Induced Changes in Physio-Biochemical Parameters of Mustard Cultivars and Their Role in Heat Stress Tolerance at the Seedling Stage

**DOI:** 10.3390/plants12061400

**Published:** 2023-03-21

**Authors:** Ashwini Sakpal, Sangita Yadav, Ravish Choudhary, Navinder Saini, Sujata Vasudev, Devendra K. Yadava, Sezai Ercişli, Romina Alina Marc, Shiv K. Yadav

**Affiliations:** 1Division of Seed Science and Technology, ICAR-Indian Agricultural Research Institute, Pusa Campus, New Delhi 110012, India; 2Division of Genetics, ICAR-Indian Agricultural Research Institute, Pusa Campus, New Delhi 110012, India; 3ICAR-Indian Council of Agricultural Research, Krishi Bhawan, New Delhi 110001, India; dkygenet@gmail.com; 4Faculty of Agriculture, Ataturk University, Erzurum 25240 and HGF Agro, Ata Teknokent, TR-25240 Erzurum, Turkey; 5Food Engineering Department, Faculty of Food Science and Technology, University of Agricultural Sciences and Veterinary Medicine, 400372 Cluj-Napoca, Romania

**Keywords:** antioxidant, heat/high-temperature stress, mustard cultivars, proline content, seedling stage, survival percentage, thermo-tolerant

## Abstract

In the era of global warming, heat stress, particularly at the seedling stage, is a major problem that affects the production and productivity of crops such as mustard that are grown in cooler climates. Nineteen mustard cultivars were exposed to contrasting temperature regimes—20 °C, 30 °C, 40 °C and a variable range of 25–40 °C—and evaluated for changes in physiological and biochemical parameters at the seedling stage to study their role in heat-stress tolerance. Exposure to heat stress showed detrimental effects on seedling growth as revealed by reduced vigor indices, survival percentages, antioxidant activity and proline content. The cultivars were grouped into tolerant, moderately tolerant and susceptible based on the survival percentage and biochemical parameters. All the conventional and three single-zero cultivars were found to be tolerant and moderately tolerant, respectively, while double-zero cultivars were reckoned to be susceptible except for two cultivars. Significant increases in proline content and catalase and peroxidase activities were found associated with thermo-tolerant cultivars. More efficient antioxidant system activity and proline accumulation were noticed in conventional along with three single-zero (PM-21, PM-22, PM-30) and two double-zero (JC-21, JC-33) cultivars that might have provided better protection to them under heat stress than the remaining one single- and nine double-zero cultivars. Tolerant cultivars also resulted in significantly higher values of most of the yield attributing traits. Heat-stress-tolerant cultivars could easily be selected based on the survival percentage, proline and antioxidants at the seedling stage and included as efficient cultivars in breeding programs.

## 1. Introduction

Food production must be doubled before 2030 in order to feed the ever-increasing world population. The food crisis has strongly been linked to the changes in global climate which are negatively impacting agricultural productivity through higher temperatures during crop production [1]. By 2050, the average temperature is expected to rise by 2 °C to 4 °C [2]. With this background, it is of absolute necessity to improve the heat-stress tolerance traits of staple crops for world food security.

High temperatures negatively impact grain development specifically after fertilization [3]. Due to the increase in heat, the growth and development during the cropping season of many of the crops are affected which results in a reduction in yields and a poor quality of seeds [4]. The most significant consequence of terminal heat stress is accelerated seed development, which leads to smaller seeds [5]. Heat stress impacts a range of physiological mechanisms involved in plant growth [6]. Stress induces inactivation of chloroplast enzymes due to the disruption of thylakoid membranes leading to the inhibition of the electron transport chain in chloroplasts and the ensuing decrease in the photosynthetic activity of plants [7,8]. The losses in chlorophyll content have been observed in many plant species under unfavorable temperature conditions [9,10,11]. Thus, the photosynthetic pigment content could also be an essential criterion for evaluating the effect of temperature stress on seed quality and yields [10]. Therefore, the examination of changes in photosynthetic activity is an important aspect to understand the consequences of heat stress in plants.

Heat stress might uncouple enzymes and metabolic pathways which cause the accumulation of unwanted and harmful reactive oxygen species (ROS) that are responsible for oxidative stress [12]. Generally, ROS damage cellular components by the oxidation of various biomolecules, e.g., DNA, lipids and proteins, which severely affect plant metabolism, growth and yield [13]. Proline is an amino acid, known to be one of the most extensively distributed amongst the compatible solutes [14], and its accumulation is positively correlated with plant stress. It is involved in the osmotic adjustment of the cytoplasm, the safeguarding of protein structures and enzyme activities against denaturation, the scavenging of hydroxyl free radicals and stabilizing cell membranes by interacting with phospholipids [15]. Thus, in order to protect themselves, plants have evolved enzymatic (CAT, POD, SOD) and non-enzymatic (glutathione, tocopherol and ascorbate) detoxification mechanisms that are widely used in the mitigation of various abiotic (heat, drought, pesticides, heavy metals) and biotic (pathogens) stresses [16].

Mustard is one of the most important edible oilseed crops worldwide. After Canada and China, India is the world’s third largest producer of mustard, accounting for about 11% of the global output [17]. It is mostly a cold-weather crop in tropical and subtropical regions [18]. More than 80% of the area in India is covered by Indian mustard (*Brassica juncea*) because of its relative tolerance to biotic and abiotic stresses in comparison with other Brassica species. The mustard crop requires an optimum average temperature of around 25 °C for the proper germination and development of seedlings. It is more sensitive to hot weather than warm-season crops [19]. High temperature (around 40 °C) severely affects embryonic cells [20] which decreases seed germination and seedling survival in the canola [21,22]. Moreover, heat has also been found to inhibit seedling development, seedling length and vigor indices in canola [22].

In India, the mustard crop constitutes about 25% of the total oilseed production under *B. juncea* species, and there is a presence of undesirable substances such as erucic acid (35–50%) in mustard oil and glucosinolate (49.9–120.3 µmol/g) in defatted seed meal which can affect human health [23]. Because of the increased global consumer awareness about the anti-nutritional factors, Indian breeders developed “canola-quality” mustard varieties: single-zero having low erucic acid content (<2%) and double-zero having both low erucic acid and glucosinolate content (<30 µmol/g defatted meal). Even though these quality mustard cultivars were nutritionally enriched, they have been reported to suffer from low vigor [24,25]. Moreover, high-temperature stress can further impact crop growth and production potential, particularly of nutritionally enriched varieties in any crops which could result in the loss of agricultural productivity. 

For a successful breeding program, it is crucial to understand the physiological and biochemical processes involved in heat-stress tolerance. Although some agronomic and genetic approaches for improving heat tolerance have been adapted in mustard, to the best of our knowledge, there is no information available on how Indian quality mustard cultivars will respond to increasing heat regimes, particularly at the early seedling stage. To the best of over knowledge, this is the first study conducted to understand the physiological and biochemical response of Indian quality mustard adaptation to heat stress during the seed germination and early seedling stages. This study would not only help in the identification of heat-tolerant Indian quality mustard cultivars but also could be helpful to those involved in the development of other field crop varieties to address the issue of hidden hunger in the world.

## 2. Results

### 2.1. Survival (Percentage)

Analysis of variance revealed significant genotypic differences and temperature treatment effects on the survival percentage. The cultivar × temperature treatment interactions were also found significant. A significantly higher survival percentage was noticed in Pusa Bold (75.7%), whereas significantly lower survival was observed in PDZ-7 (45.1%). Depending upon the values obtained from the survival percentage, the cultivars were grouped into tolerant, moderately tolerant and susceptible cultivars. The highest range of the survival percentage (61–76%) was observed in tolerant cultivars followed by moderately tolerant cultivars (55–60%), and the lowest survival percentage was found in susceptible cultivars (45–50%).

Survival percentage differed with the application of heat/temperature treatments. It was found that the highest survival percentage (32.3–75.1%) was recorded at 30 °C, while it was found lowest (15.2–48.3%) at 40 °C. Among the interaction effects of temperature treatments and groups of cultivars, Pusa Bold showed a maximum survival percentage (75.1%) at 30 °C, while a minimum survival percentage was observed in PDZ-7 (32.3%) at 30 °C (Table 1).

Among the groups, the seedling survival percentage in tolerant cultivars was 46.3% at 40 °C, 55.5% at a variable temperature range of 25–40 °C and 65.0% at 30 °C. Meanwhile, in moderately tolerant cultivars, it was 30.7% at 40 °C, 38.9% at a variable temperature range of 25–40 °C and 58.9% at 30 °C. The minimum survival was observed in the case of susceptible cultivars, and it was 18.4% at 40 °C, 27.3% at a variable temperature range of 25–40 °C and 40.9% at 30 °C (Figure 1). Thus, it was recorded that the survival percentage was higher in tolerant (group III) followed by moderately tolerant cultivars (group II) compared to susceptible cultivars (group I).

Manhattan distance-based hierarchical clustering on biochemical parameters was also found to organize the mustard cultivars into three major groups, i.e., groups I, II and III, similar to those indicated by the survival percentage (Figure 2). The maximum chlorophyll content (661, 567 and 484 µg/g FW), catalase activity (35.6, 27.6 and 19.4µmol/g FW), peroxidase activity (603, 458 and 307µmol/g FW), superoxide dismutase (SOD) (375, 351 and 292 U/g FW) and proline content (9.5, 7.2 and 4.8 µmol/g FW) in group III followed by group II and minimum were found in group I, respectively, under different treatments (Figure 3).

Group I contains ten cultivars which showed the lowest chlorophyll content, antioxidant enzyme activities and proline content due to their susceptible behavior against heat stress. Among the ten cultivars, nine cultivars (PDZ-1, PDZ-3, PDZ-4, PDZ-5, PDZ-7, PDZ-8, PDZ-11, PDZ-12 and Heera) were double-zero quality mustard, and the remaining one was the Pusa Karishma single-zero quality mustard cultivar. Group II demonstrated a lower decrease in biochemical parameters at higher temperatures and thus was characterized as moderately tolerant to heat stress. It comprised four cultivars (Figure 2): JC-21 (double-zero quality mustard), PM-21 and PM-22 (single-zero quality mustard) and Navgold (conventional cultivar). 

Group III revealed a marked increase in chlorophyll content, antioxidant enzyme activities and proline content under higher temperatures, and therefore, the cluster was classified as extremely tolerant to heat stress. It comprises five cultivars (Figure 2), namely, Pusa Bold, Pusa Vijay and BEC-144 (conventional type), PM-30 (single-zero quality mustard) and JC-33 (double-zero quality mustard).

### 2.2. Germination Percentage and Mean Germination Time (Days)

The germination percentage was significantly influenced by the interaction between temperature and groups of cultivars. Among the groups, the germination in tolerant cultivars was 42.3% at 40 °C followed by 54.1% at a variable temperature range of 25–40 °C, and the highest germination was 62.9% found at 30 °C. Meanwhile, in the case of moderately tolerant cultivars, the lowest germination was 38.6% found at 40 °C followed by 51.7% at a variable temperature range of 25–40 °C, and the highest germination was 63.7% observed at 30 °C. In the case of susceptible cultivars, the lowest germination was 33.1% recorded at 40 °C followed by 42.6% at a variable temperature range of 25–40 °C, and the highest germination was 56.4% found at 30 °C (Figure 4A). 

Overall, the results revealed a decline in germination percentages among all the groups during heat treatments as compared to control. The decline in the germination percentage was more significant and prominent in susceptible cultivars (group I) followed by moderately tolerant and tolerant cultivars (group II and III), respectively. It indicated that there were cultivar differences in the germination percentage with respect to their ability to tolerate heat stress.

Mean germination time (MGT) is regarded as a measure of the rate and time spread of germination. Significant differences in MGT were also observed with respect to the interaction effect of treatments and groups of cultivars. Among the three groups, tolerant cultivars (group III) required a minimum of 3.65 days at 30 °C followed by 3.74days when exposed to a variable temperature range of 25–40 °C and 4.67 days at 40 °C, whereas susceptible cultivars (group I) required a maximum of 3.67 days at 30 °C followed by 3.89 days at variable temperature and 4.87 days at 40 °C. The differences in mean germination time were found to be non-significant among the three cultivar groups (Figure 4B).

### 2.3. Seedling Vigor Index II (SVI-II)

The result for SVI-II was computed based on the germination percentage and the seedling’s dry matter. The seedling vigor index II showed significant differences in the interaction effect of temperature treatments and groups of cultivars. Among the different groups, tolerant cultivars (group III) recorded maximum SVI-II (1.55, 1.31 and 1.14), while susceptible cultivars (group I) recorded minimum SVI-II (1.26, 1.08 and 0.86) at 30 °C, a variable temperature range of 25–40 °C and 40 °C, respectively (Figure 5).

### 2.4. Biochemical Parameters

Biochemical parameters such as chlorophyll content, antioxidant enzyme activity and proline content showed significant differences in cultivars. Among the cultivars, the highest significant chlorophyll content (636µg/g FW), catalase activity (33.4 µmol/g FW), peroxidase activity (558 µmol/g FW), superoxide dismutase activity (360 U/g FW) and proline content (8.61 µmol/g FW) were noticed in conventional mustard cultivars followed by single-zero cultivars. Meanwhile, double-zero mustard cultivars had the lowest significant chlorophyll content (489 µg/g FW), catalase activity (21.6 µmol/g FW), peroxidase activity (348 µmol/g FW), superoxide dismutase activity (304 U/g FW) and proline content (5.29 µmol/g FW). Thus, the most efficient antioxidant system and proline content were observed in conventional cultivars followed by single-zero cultivars in relation to biochemical parameters compared to the double-zero mustard cultivars (Table 2).

Based on the analysis of variances, the main effect of individual cultivars with biochemical parameters such as chlorophyll content, catalase, peroxidase, superoxide dismutase activity and proline content were significantly influenced. Among the cultivars, the highest significant chlorophyll content (749 µg/g FW) was observed in PM 30 (single-zero mustard), which was at par with Pusa Bold (conventional mustard) (739 µg/g FW), and the lowest chlorophyll content (395 µg/g FW) was found in PDZ-5 (double-zero mustard). Significantly higher catalase (37.3 µmol/g FW) and peroxidase (633 µmol/g FW) activities were noticed in Pusa Bold (conventional mustard), whereas significantly lower catalase activity (16.0 µmol/g FW) was observed in PDZ-7 (double-zero mustard) and peroxidase activity (265 µmol/g FW) was found in the Heera cultivar (double-zero mustard). The highest SOD (404 U/g FW) was recorded in Pusa Bold (conventional mustard), and the lowest (276 U/g FW) was in PDZ-5 (double-zero mustard). The maximum proline content (10.39 µmol/g FW) was observed in PM-30 (single-zero mustard), and the minimum was observed in PDZ-4 (double-zero mustard) (4.45 µmol/g FW) (Table 2).

#### 2.4.1. Chlorophyll Content (µg/g FW) 

Significant differences were observed in chlorophyll content for the interaction effect of treatments and groups of cultivars. Among the three groups, tolerant cultivars showed the highest chlorophyll content in all the treatments. Chlorophyll content of 749 µg/g FW was found at 30 °C, 543 µg/g FW at a variable temperature range of 25–40 °Cand 395 µg/g FW at 40 °C. On the other hand, susceptible cultivars observed the lowest chlorophyll content at all treatments: 552 µg/g FW, 378 µg/g FW and 288 µg/g FW, respectively. Thus, the result showed that tolerant cultivars (group III) maintained high chlorophyll content even at higher heat-stress conditions (Figure 6).

#### 2.4.2. Antioxidant Enzyme Activities

##### Catalase and Peroxidase Activity (µmol/g FW)

Antioxidant enzymes play a very important role under stress conditions. The effect of high temperature was evident in antioxidant enzymes which differed significantly for the interaction between treatments and groups of cultivars. Catalase and peroxidase are important enzymes to detoxify hydrogen peroxide (reactive oxygen species). Among the different groups, tolerant cultivars recorded maximum activity of 30.6, 38.3 and 49.5 µmol/g FW by catalase enzyme and activity of 569, 671 and 725 µmol/g FW by peroxidase enzyme at 30 °C, a variable temperature range of 25–40 °C and 40 °C, respectively. Susceptible cultivars recorded lower activities of 15.4, 22.3 and 28.8 µmol/g FW by catalase enzyme and activity of 258,352 and 421 µmol/g FW by peroxidase enzymes at 30 °C, a variable temperature range of 25–40 °C and 40 °C, respectively. The result revealed that tolerant cultivars had maximum catalase and peroxidase activity compared to susceptible cultivars during heat stress (Figure 7A,B).

##### Superoxide Dismutase Activity (U/g FW) 

Among the different groups, group III containing tolerant cultivars recorded maximum superoxide dismutase (SOD) activity of 380 U/g FW at 30 °C followed by 312 U/g FW at a variable temperature range of 25–40 °C and 260 U/g FW at 40 °C, and minimum SOD of 320 U/g FW was observed in susceptible cultivars (group I) at 30 °C followed by 232 U/g FW at a variable temperature range of 25–40 °C and 179 U/g FW at 40 °C (Figure 8A).

#### 2.4.3. Proline Content (µmol/g FW)

Proline content can increase upon the exposure of plants to any type of stress. The interaction between groups of cultivars and treatments showed significant differences in proline content. Among the different groups of cultivars, tolerant cultivars recorded maximum proline content of 9.1µmol/g FW at 30 °C followed by 11.4 µmol/g FW at a variable temperature range of 25–40 °C and 15.7 µmol/g FW at 40 °C, while minimum proline content was recorded in susceptible cultivars at 30 °C (3.7 µmol/g FW), followed by the variable temperature range of 25–40 °C (5.9 µmol/g FW) and 40 °C (8.4 µmol/g FW). Thus, the results suggested that among the three groups of cultivars, tolerant cultivars showed the highest proline content, while it was lowest in susceptible cultivars under heat-stress conditions (Figure 8B). 

### 2.5. Correlation Analysis between Different Parameters

Correlation among the germination percentage, MGT, survival percentage, seedling vigor index II, chlorophyll content, proline content, catalase, peroxidase and SOD activity was measured in 19 mustard cultivars under contrasting temperatures of 20 °C, 30 °C, a variable temperature range of 25–40 °C and 40 °C (Table 3). Both the positive and negative correlations were found to be statistically significant at *p* <0.01. The germination percentage showed a significant positive correlation with seedling vigor index II (r = 0.79). Since SVI-II is the manifestation of germination and dry weight, it was an obvious positive relationship. A significant negative correlation (r = −0.73) was observed between the germination percentage and MGT and also between MGT and SVI-II (r = −0.66). These correlations revealed that more time is required by the seedling to emerge and to overcome stress under high temperatures. In the present study, the levels of proline content measured showed a strong negative correlation with the survival percentage of plants (r= −0.54), whereas catalase and peroxidase activities showed a positive correlation with proline content and a negative correlation with SVI-II, SOD and chlorophyll content. The superoxide dismutase (SOD) enzyme protects cells from radical attacks. So, SOD activity was noticed positively correlated to the survival percentage (r = 0.92) and chlorophyll content (r = 0.86).

### 2.6. Seed Yield Attributing Traits

Seed yield attributing characteristics, such as the number of primary as well as secondary branches, main shoot length, number of siliquae on the main shoot, siliqua length and seed yield per plant, showed significant differences among groups of cultivars (Table 4), whereas the number of seeds per siliqua and siliqua density on the main shoot showed no significant differences. Among the groups of cultivars, the highest significant number of primary (10.8) as well as secondary branches (8.5), main shoot length (69.5 cm), number of siliquae on the main shoot (39.3), siliqua length (5.01 cm) and seed yield per plant (40.9 g) were observed in tolerant cultivars, whereas susceptible cultivars had the lowest significant number of primary (8.5) as well as secondary branches (22.8), main shoot length (50.6 cm), number of siliquae on the main shoot (31.6), siliqua length (3.37 cm) and seed yield per plant (31.9 g). Thus, tolerant cultivars (group III) were found to be better performers with respect to the above-mentioned parameters than the susceptible cultivars (group I).

## 3. Discussion

Indian mustard is widely grown under different agro-ecological conditions [26]. The seedling as well as flowering stages are the most sensitive to heat stress [27]. Moreover, the increasing global air temperature at the rate of 0.2 °C per decade is raising apprehension regarding crop productivity and food security [28]. This global hike in temperature is becoming the critical limiting factor for plant growth, development and yield attributes. Impediment to seed germination and field emergence is the first effect of heat stress which is observed in many crops [29,30].

Seed germination and early seedling growth are the most sensitive stages for plant stand establishment under extreme environmental stresses [31]. Increased temperature stress has been found to have a strong negative effect on seed germination potential resulting in poor germination [32]. The germination percentage of all the groups of cultivars was significantly reduced with increasing heat stress. The reduction in germination percentage was more prominent in susceptible cultivars. Hence, the results suggested that there were genotypic differences in germination with respect to their ability to tolerate high temperatures. Zhang et al. [22] also reported significant differences in the germination percentage of different canola varieties. 

Efficient seed germination and early, rapid and uniform seedling emergence are determinants of effective plant stand in the field [33]. Mean germination time is one of the physiological indicators of seed vigor [34]. An increase of 2.2% in MGT was found at 30 °C, 8.4% at a variable temperature range of 25–40 °C and 34.7% at 40 °C over the control. The increase in MGT during heat stress, therefore, results in lower and non-uniform seed germination and seedling establishment. The suppression of seed germination under heat stress could be due to an increase in ABA levels or a decrease in GA biosynthesis as reported by Toh et al. [35] in *Arabidopsis*. 

Seed vigor is a complex physiological trait that determines duration and successful seedling establishment under a wide range of environmental conditions. It encompasses germination speed, seedling growth, early stress tolerance and seed longevity. Variable detrimental effects on seedling vigor index II were observed under various treatments. The reduction in vigor could be due to the enhanced accumulation of stress-induced toxic compounds such as reactive oxygen species (ROS) in cells [36]. Elevated temperatures have also been shown to influence seed germinability, in terms of vigor in rice [37]. Cellular homeostasis is essential for the normal physiological and biochemical functioning of the cell. However, under heat stress, this co-ordination between multiple pathways occurring in different organelles is disrupted because of the accumulation of toxic compounds such as ROS. Such disruptions could severely affect the function and structure of chlorophyll resulting in membrane damage with reduced photosynthesis [38]. Hu et al. [39] reported that a dramatic increase in chlorophyllase and chlorophyll-degrading peroxidase resulted in a serious depletion of chlorophyll levels at high temperatures. A similar decline in chlorophyll content was observed in canola [22] and alfalfa [40] upon exposure to heat stresses. However, tolerant cultivars seem to have better protective mechanisms against photo-oxidation by heat stress; hence, tolerant cultivars show higher chlorophyll content than their sensitive counterparts. Similar results have been reported in rice and sorghum [41,42]. The most damaging consequence of these perturbations is the overproduction of deleterious ROS damaging the macromolecules and subcellular components [43]. The seedlings, on other hand, have evolved a protective antioxidative enzymatic, as well as non-enzymatic, antioxidant defense system including POD, SOD and CAT to lessen the harm caused by ROS [44,45]. The ameliorating effects of antioxidants on heat stress were also reported in moth beans [46] and wheat crops [47]. Higher CAT and POD activity at high-temperature regimes suggests their higher capacity to neutralize deleterious hydrogen peroxide, thereby minimizing damage to cells and resulting in enhanced tolerance of high temperature [48,49]. Thus, higher activities of antioxidant enzymes in tolerant cultivars under heat-stress conditions might have helped mustard seedlings to cope with oxidative damage. To manage the challenges due to heat stress, plants have evolved many mechanisms. One of them is the accumulation of osmolytes such as proline in the cytoplasm to maintain the plant water potential during heat stress. It also plays the role of molecular chaperons which preserve the structure of proteins, stabilize sub-cellular membrane structures [50], scavenge free radicals and maintain cell redox status [51]. Proline has been reported to increase due to high temperatures in different plant species [52,53,54]. The heat-tolerant cultivars possessed greater proline content than heat-sensitive cultivars. This indicates that high temperature showed many more adverse effects on seedlings; hence, proline acts as a reliable indicator of the environmental stress imposed on the plants by heat stress [15]. The study also revealed that tolerant cultivars had significantly higher values for most of the yield attributing characteristics that could be attributed to the genetic makeup of these cultivars and their higher capacity to utilize the photosynthates. The efficient utilization of reserves could result in a higher number of branches/plants leading to elevated dry matter production and ultimately resulting in better yields [53].

Overall, susceptible cultivars showed a drastic reduction in germination percentage and vigor. They were also found to have high MGT and lower antioxidant, chlorophyll and proline content. Thus, the reduced antioxidant defense, proline content and chlorophyll content indicate that these seedlings have undergone considerable oxidative damage during heat stress. It is due to this early damage that the seedlings are unable to recover in phase II of seed germination. Therefore, high temperature prevailing at the time of sowing reduces seed germination and causes seedling mortality in Indian quality mustard and could cause similar reductions in the germination of other crops. This results in poor crop stand establishment, leading to reduced yields [55].

However, to accelerate the breeding program for heat-stress tolerance, “early selection” of heat-tolerant cultivars is essential. This will not only reduce the workload but also save time for the selection process. The grouping of cultivars based on the survival percentage is a rapid and inexpensive method for screening cultivars for heat tolerance. Under optimum conditions (20 °C), all groups of cultivars showed 100% survival, but as the temperature was increased, the survival percentage decreased. Under high temperature, higher survival was recorded in tolerant and moderately tolerant cultivars (all conventional, single-zero, except for Pusa Karishma and two double-zero cultivars, JC-21 and JC-33) as compared to susceptible cultivars (nine double-zero cultivars). The higher survival percentage of seedlings reflects the recovery potential within the cultivars under heat-stress conditions. The tolerance capacity of cultivars is the outcome of physiological and biochemical changes under stress conditions [56]. These results revealed the limited ability of susceptible cultivars to adapt to heat-stress conditions during the germination stage. The susceptible cultivars have been reported sensitive to high temperature and require more ideal temperatures for cultivation [27].

Heat stress greatly reduced the germination of seeds of Indian mustard. Based on the survival percentage, it was observed that not only conventional cultivars but also some quality mustard cultivars also showed resistance to heat-stress (high-temperature) conditions. The survival percentage has also been reported as a selection criterion for thermal stress in rice [57]. Among the quality mustards, double-zero quality mustard, i.e., JC-33 was found tolerant, and JC-21 was moderately tolerant, while single-zero quality mustard cultivars PM-22 and PM-30 were found tolerant, and the PM-21 cultivar was found moderately tolerant. Thus, these quality mustard cultivars, which otherwise suffer from low seeding vigor, can be successfully grown in any area having relatively high temperatures. The present study is the first report on the effect of heat stress on quality Indian mustard, and the survival percentage can be used as a reliable trait to evaluate the heat tolerance of *Brassica* spp.

## 4. Materials and Methods

### 4.1. Experimental Materials

Pure seeds of 19 Indian mustard cultivars including conventional and quality mustard were collected from the Division of Genetics, Indian Council of Agricultural Research (ICAR)—Indian Agricultural Research Institute (IARI), New Delhi (India). The procured cultivars belong to the following categories: 4 conventional (Pusa Vijay, Pusa Bold, Navgold and BEC-144), 4single-zero (PM-21, PM-22, PM-30 and Pusa Karishma) and 11 double-zero (Heera, JC-21, JC-33, PDZ-1, PDZ-3, PDZ-4, PDZ-5, PDZ-7, PDZ-8, PDZ-11 and PDZ-12). Among the quality mustard cultivars, single-zero mustard has low erucic acid content (<2%), and double-zero mustard has both low erucic acid and low glucosinolate content (approximately <30 µmol/g defatted meal), in contrast to conventional mustard, which has both non-nutritional factors in high quantities. Uniform-sized seeds were selected, and 200 g of each cultivar was surface-sterilized with 1.0% (*w*/*v*) sodium hypochlorite for 15 min and subsequently rinsed six times with distilled water before germination. The sterilized seeds were first wiped with blotter paper and placed in 15 cm Petri dishes to dry at ambient temperature under a fan in shade.

Completely dried seeds of all the cultivars were used for further experiments under four different temperature regimes. The trials for heat stress were conducted in four sets of temperatures: T1, the control set, was kept at an ambient temperature of 20 °C with a relative humidity of 95% (walk-in germinator). The other two sets were kept in the growth chamber at the Division. The T2 set was kept at a constant temperature of 30 °C with a relative humidity of 75%, and the T4set was kept at a constant temperature of 40 °C with a relative humidity of 65%. The remaining set (T3) was kept in the growth chamber with a sequence of specific temperatures along with durations similar to those that are experienced in actual field conditions during the seedling establishment of Indian mustard as shown in (Figure 9). Optimal conditions with respect to light and moisture were maintained for proper germination and development of seedlings. The photoperiod was set at 12 h a day, and the minimum photosynthetic photon flux density (PPFD) was maintained at 440 μmol m^−2^ s^−1^ during the daytime. The germination percentage and survival percentage of seedlings were determined. The chlorophyll content, free proline concentration and activities of superoxide dismutase (SOD), catalase (CAT) and peroxidase (POD) enzymes were also measured to evaluate the physiological and biochemical responses to high-temperature stress.

### 4.2. Methodology

#### 4.2.1. Survival Percentage

Evaluation of heat-stress tolerance of mustard cultivars was undertaken in terms of survival percentage after exposure to various temperature treatments. Six seedling/plug trays with 50 wells for each treatment combination were filled with sandy loam soil and farm yard manure (6:1 ratio) for sowing the seeds, and to each well, measured amount of water was added to bring the moisture levels to field capacity. On the third day of germination, 5 mL/well of water was sprayed, and each subsequent day after that, the seedlings received the same amount of water. Once the sterilized seed of each cultivar was sown in each well of the six trays, it was subjected to four temperature stresses as mentioned above, thus making three replications of 100 seeds in each treatment combination. The recording of observations for survival percentage started after 5th day (the day of first count) and continued until9th day from the date of sowing, and survival percentage was calculated using the following formula:(1)$$\mathrm{Survival}\;\left(\%\right)=\frac{\mathrm{Number}\;\mathrm{of}\;\mathrm{seedlings}\;\mathrm{survived}\;\mathrm{after}\;9\mathrm{th}\;\mathrm{days}}{\mathrm{Total}\;\mathrm{number}\;\mathrm{of}\;\mathrm{seedlings}\;\mathrm{germinated}\;\mathrm{after}\;5\mathrm{th}\;\mathrm{day}}\times 100$$

#### 4.2.2. Germination Percentage and Mean Germination Time (d)

The seedlings germinated in the plug trays put in walk-in germinator were categorized into normal and abnormal seedlings. A seedling that possessed all the essential structures that indicate its ability to develop into mature plant under favorable field conditions was considered to be normal seedling, whereas a seedling that didnot have all essential structures or was damaged, deformed or decayed, which prevents normal development, was considered an abnormal seedling [58]. The number of normal seedlings was counted on the 7th day of sowing in three replications of all treatment combinations. The average of the number of normal seedlings germinated gave final germination and was expressed as a percentage. The number of seeds germinated in the plug trays put in walk-in germinator and growth chambers was counted daily in all 3 replications for all treatment combinations. The mean germination time (MGT) was calculated following the method of Nicholas and Heydecker [59]. The number of seeds germinated each day was recorded up to the final day (7th) of the count. Results were calculated with the mean of replicates using the following formula:(2)$$\mathrm{MGT}=\frac{\sum \mathrm{Dn}}{\sum n}$$
where *n* = number of seeds, germinated on day D, and D = number of days counted from the beginning of germination.

#### 4.2.3. Seedling Dry Weight (g)

For recording seedling dry weight, the fresh weight (FW) (g) of 10 seedlings was taken. After recording the FW, the seedlings were placed on wax paper and allowed to dry in a hot-air oven at 70 ± 1°C for 48 h. The seedling dry weights (SDW) were measured after cooling for 30 min in a desiccator with silica gel. The values of SDW expressed in grams (g per 10 seedlings) were used for calculations of seedling vigor index (SVI) II.

#### 4.2.4. Seedling Vigor Index II

Seedling vigor index II was calculated using the formula below, following the method of Abdul-Baki and Anderson [60]: Seedling vigor index-II (SVI-II) = Germination (%) × Seedling dry weight (g).

#### 4.2.5. Chlorophyll Content (µg/g FW)

Chlorophyll content was determined using 100–150 mg of seedlings from all the treatments in three replications. Chlorophyll content was extracted with 10 mL of 80% (v/v) chilled acetone, and the absorbance for Chl-a and Chl-b was recorded at 645 nm and 663 nm, respectively, using a spectrophotometer (UV-1800, Shimadzu Corporation, Kyoto, Japan), and the chlorophyll content (µg/g FW) was calculated using the formula [(20.2 × A645) + (8.02 × A663)]/1000 × W× V, where W is the seedling weight, and V is the extraction volume [61].

#### 4.2.6. Antioxidant Enzyme Activities

Fresh whole seedlings were ground in an ice-cold mortar with 8 mL of 50 mM cold phosphate buffer (pH 7) and centrifuged at 15,000 rpm for 20 min at 4 °C. A supernatant was used for the determining activities of the following enzymes [62]. 

##### Catalase (CAT) Assay (µmol/g FW) 

CAT activity was determined spectrophotometrically by its ability to catalyze the decomposition of H_2_O_2_ at 240 nm. The reaction was initiated by the addition of the above supernatant to the reaction mixture containing 50 mM phosphate buffer (pH 7.0) with 10 mM H_2_O_2_. The changes in absorbance were estimated at 240 nm at15 s intervals for 3 min [63].

##### Peroxidase (POD) Assay (µmol/g FW) 

POD activity was determined by its ability to oxidize guaiacol at 470 nm. The reaction was performed in a 3 mL solution. The above supernatant was added to phosphate buffer (100 mM, pH 6.0) containing 96 mM guaiacol and 12 mM H_2_O_2_ to start the reaction. Finally, the changes in absorbance were recorded at 470 nm at 15 s intervals for 3 min [64].

##### Superoxide Dismutase Activity (SOD) Assay (U/g FW) 

SOD activity was assayed by inhibition of the photochemical reduction of nitro blue tetrazolium (NBT) as described by Donahue et al. [65]. Three-milliliter reaction buffer contained 50 mM phosphate buffer (pH 7.8), 0.1 mM EDTA, 13 mM methionine, 75 µm NBT, 2 µM riboflavin and 50 µL of the above supernatant. Riboflavin was added last, and reaction was initiated by placing the tubes under 15-watt fluorescent lamps. The reaction was terminated after 10 min by switching off the light when the tubes were covered with a black cloth. Non-illuminated and illuminated reactions without supernatant served as calibration standards. The reaction product was measured at 560 nm. SOD activity was expressed as U/g FW, where one unit of SOD activity was defined as the amount of enzyme that caused 50% inhibition of the photochemical reduction of NBT.

#### 4.2.7. Proline Content (µmol/g FW)

The whole seedlings of weight 150–200 mg were taken for determination of proline content by adopting the method of Bates [66] with some modifications. Samples were extracted with sulphosalicylic acid (3%). In the extract, an equal volume of glacial acetic acid and ninhydrin solutions were added. Thereafter, they were kept in a boiling water bath set at 100 °C for 1 h, and the reaction was ended by putting them in an ice bath. Then, 5 mL of toluene was added to the reaction mixture and mixed vigorously with the vortex for 15–20 s. After the mixture was warmed back to room temperature, the supernatant was collected to read the absorbance at 520 nm using a spectrophotometer (UV-1800, Shimadzu Corporation, Kyoto, Japan). The proline content was estimated using a standard concentration curve and calculated on a fresh weight basis (µmol/g FW).

#### 4.2.8. Yield Parameters

The nineteen cultivars were evaluated for performance in field during the rabi seasons of 2020–2021 and 2021–2022. Fifty seeds of each cultivar were sown in the plot size of 2.4 m × 3 m with three replications each, and recommended package of practices was followed to raise healthy crops. The mean weather data (Table 5) of the entire crop growth periods were collected from agro-met observatory of the Division of Agricultural Physics, ICAR-IARI, New Delhi (India).

The observations were recorded in 5 individual plants from each cultivar for the following attributes.

##### No. of Primary Branches per Plant 

The number of primary branches emerging directly from the main shoot of the plant was recorded in 5 individual plants from each type of cultivar. 

##### No. of Secondary Branches per Plant 

The number of siliquae bearing branches emerging from the primary branches wascounted at maturity and was recorded in 5 individual plants from each cultivar. 

##### Main Shoot Length (cm)

The main shoot length from all five individual plants from each type of cultivar was measured.

##### Siliqua Length (cm)

Five siliquae from the main raceme of 5 individual plants from each type of cultivar were taken randomly, and after that, the siliqua length was measured and averaged. 

##### Number of Siliquae on the Main Shoot 

Number of siliquae present on the main shoot of the selected plants was counted and averaged. 

##### Siliqua Density on Main Shoot 

Siliqua density was worked out by dividing the number of siliquae on main shoot with main shoot length in all five individual plants of each type of cultivar and averaged. 

##### The Number of Seeds per Siliqua 

Five siliquae from main raceme of each plant were taken randomly, and after that, the number of seeds was counted and averaged. 

##### Seed Yield per Plant (g) 

Five tagged plants of each type of cultivar were threshed separately, cleaned, weighed and averaged.

### 4.3. Experimental Design and Statistical Analyses

The lab and field data of the experiments were collected and arranged in a factorial completely randomized design with four levels of Factor 1 (temperature) and 19 levels of Factor 2 (cultivars) and a factorial randomized block design. Three replications for all the treatment combinations were applied in which 100 seeds per replication were used. Significant differences between means of treatments, cultivars and interactions were calculated using the least significant difference to compare the means using Tukey’s test at *p* ≤ 0.05, and correlation coefficients (r) were also determined using SPSS 16.0, Chicago, IL, USA. Analysis of variance (ANOVA) from the data was employed to compute the variable effects of the factors and their interaction. To estimate similarities among cultivars for survival under heat stress and biochemical parameters, cluster analysis was performed using Numerical Taxonomy and Multivariate Analysis System (NTSYS). The dissimilarity matrix was used to construct the dendrogram by the unweighted pair group method for the arithmetic mean (UPGMA) based on Sequential Agglomerative Hierarchical and Nested (SAHN) clustering [67].

## 5. Conclusions

The present study on nineteen mustard cultivars revealed that high-temperature stress caused some level of injuries at the seedling stage as well as triggered some defense mechanisms to prevent those injuries. The mustard cultivars responded differently to different temperature conditions. The tolerant cultivars showed high antioxidant enzyme activity and proline content to counteract the deleterious effect of heat stress thus resulting in better field performance. On the other hand, susceptible cultivars in group I showed fewer protective mechanisms under heat stress. The present study thus provides a simple, efficient and cost-effective method for screening, based on the survival percentage of cultivars, to help early selection of heat-tolerant cultivars.

## Figures and Tables

**Figure 1 plants-12-01400-f001:**
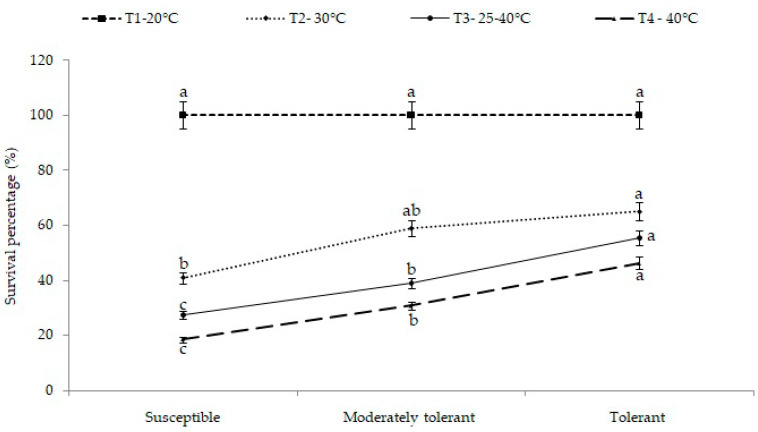
Grouping of cultivars into tolerant, moderately tolerant and susceptible sets based on survival percentage under different temperatures. Different letters on SEM (±) bars depict significant differences (*p* < 0.05), where cases of means are three (*n* = 3).

**Figure 2 plants-12-01400-f002:**
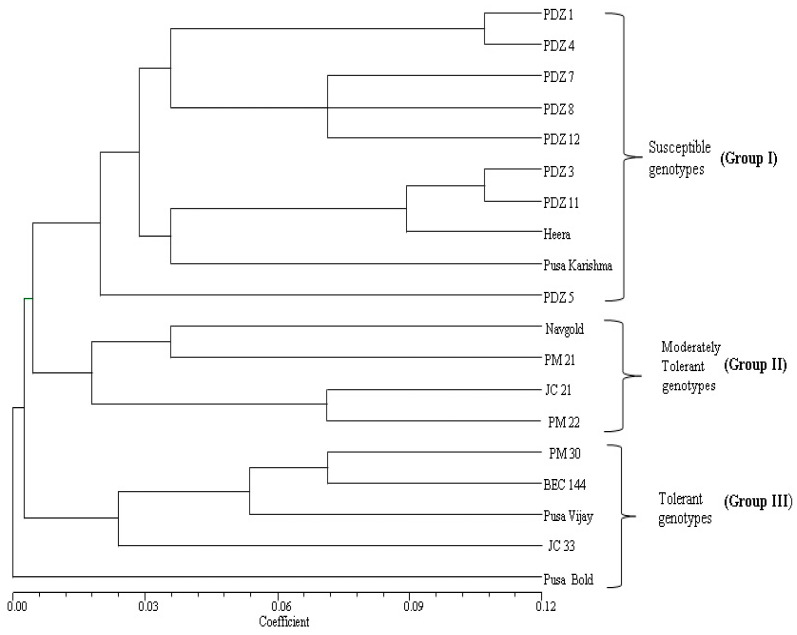
UPGMA is based on biochemical parameters of 19 mustard cultivars grown under contrasting temperatures of 20 °C, 30 °C, 25–40 °C and 40 °C.

**Figure 3 plants-12-01400-f003:**
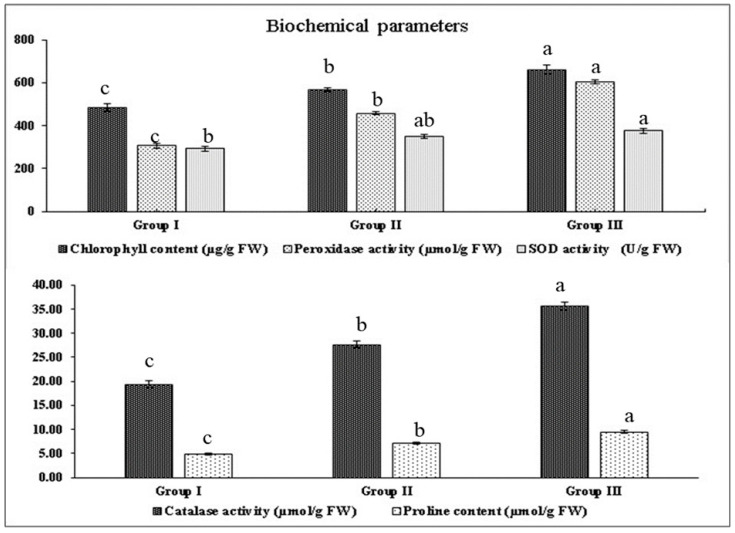
Grouping of cultivars based upon mean contents/activities of biochemical parameters under different temperatures. Different letters on SEM (±) bars depict significant differences (*p* < 0.05), where cases of means are three (*n* = 3).

**Figure 4 plants-12-01400-f004:**
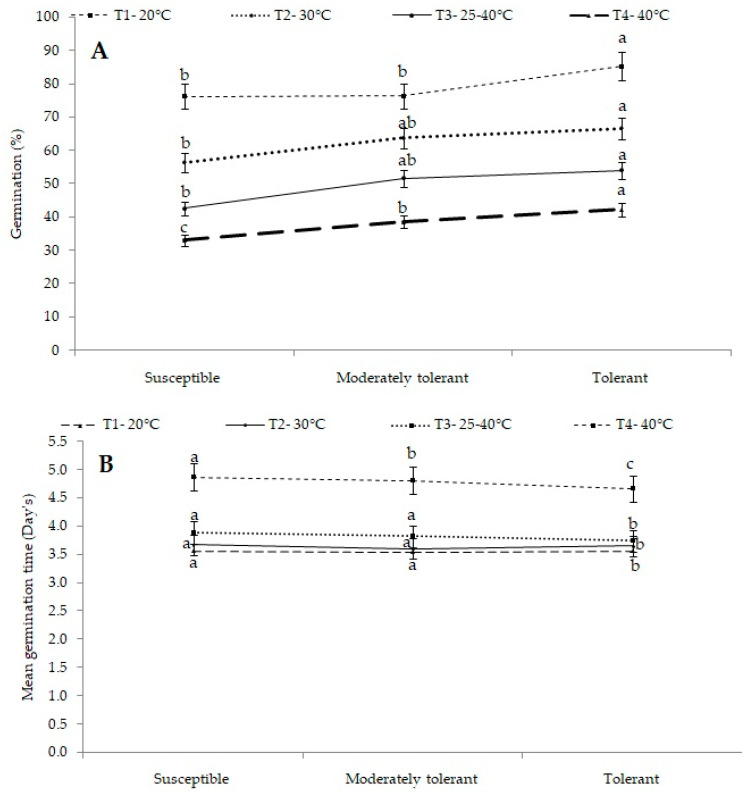
(**A**) Mean germination percentage and (**B**) mean germination time (d) of different groups of cultivars under different treatments. Different letters on SEM (±) bars depict significant differences (*p* < 0.05), where cases of means are three (*n* = 3).

**Figure 5 plants-12-01400-f005:**
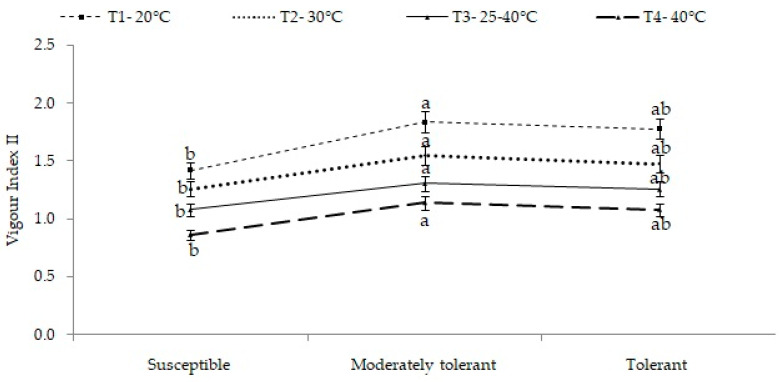
Mean seedling vigor index II of different groups of cultivars under different treatments. Different letters on SEM (±) bars depict significant differences (*p* < 0.05), where cases of means are three (*n* = 3).

**Figure 6 plants-12-01400-f006:**
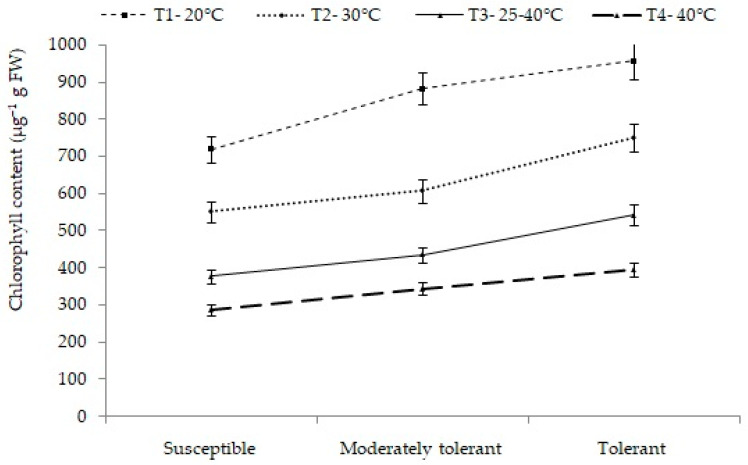
Mean chlorophyll content (µg/g FW) of different groups of cultivars under different treatments. Different letters on SEM(±) bars depict significant differences (*p* < 0.05), where cases of means are three (*n* = 3).

**Figure 7 plants-12-01400-f007:**
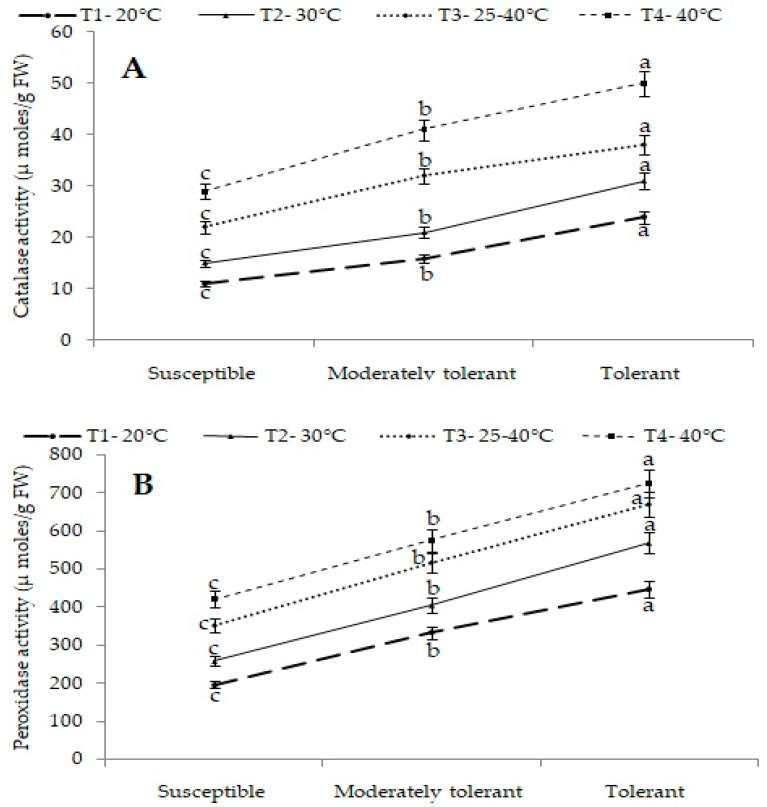
(**A**) Mean catalase activity (µmol/g FW) and (**B**) mean peroxidase activity (µmol/g FW) of different groups of cultivars under different treatments. Different letters on SEM(±) bars depict significant differences (*p* < 0.05), where cases of means are three (*n* = 3).

**Figure 8 plants-12-01400-f008:**
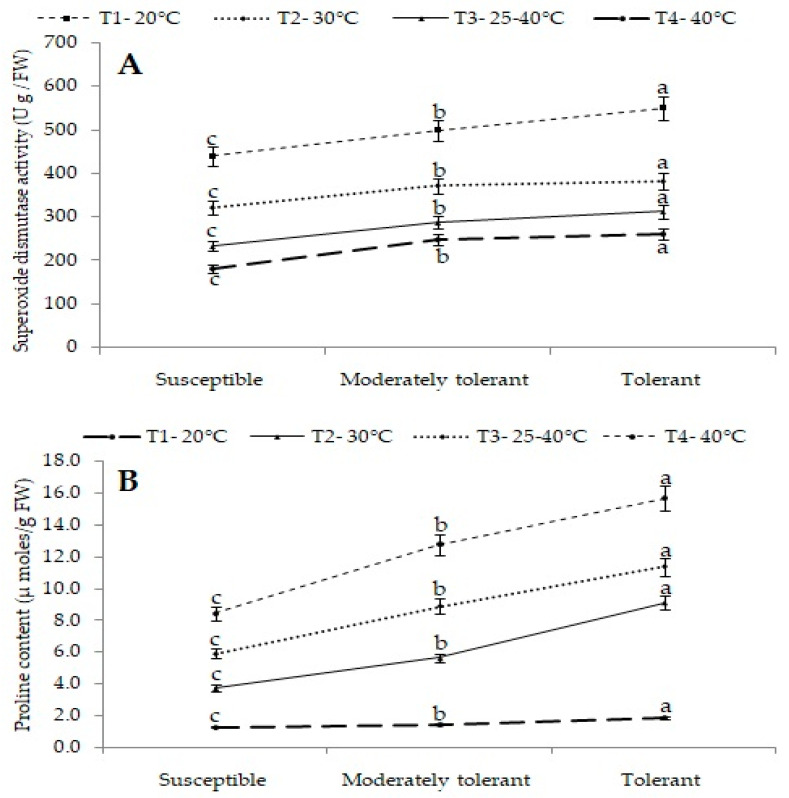
(**A**) Mean superoxide dismutase activity (U/g FW) and (**B**) mean proline content (µmol/g FW) of different groups of cultivars under different treatments. Different letters on SEM (±) bars depict significant differences (*p* < 0.05), where cases of means are three (*n* = 3).

**Figure 9 plants-12-01400-f009:**
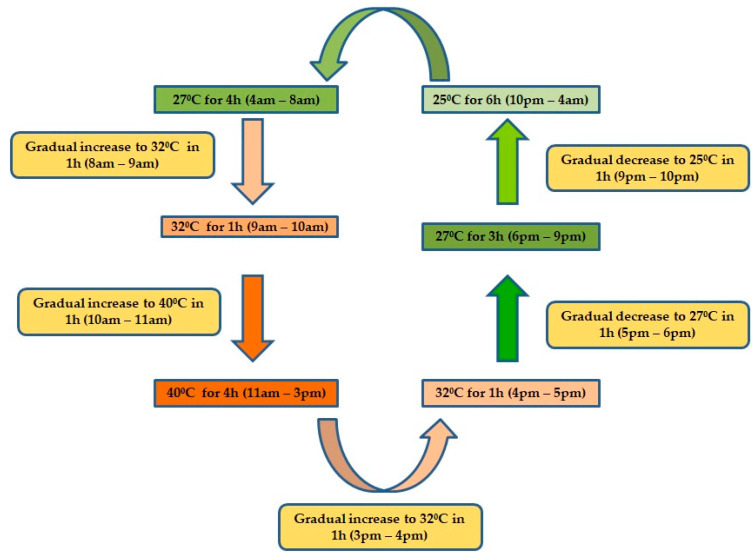
Flow chart shows the specific temperatures and durations maintained for T3 treatment in growth chamber to mimic actual field conditions.

**Table 1 plants-12-01400-t001:** Interaction effects between cultivars and temperature treatments on mean survival percentage.

		Susceptible Cultivars									Moderately Tolerant Cultivars				Tolerant Cultivars					
Temperatures	PDZ-1	PDZ-3	PDZ-4	PDZ-5	PDZ-7	PDZ-8	PDZ-11	PDZ-12	Pusa Karishma	Heera	JC-21	PM-21	PM-22	Navgold	PM-30	Pusa Bold	Pusa Vijay	JC-33	BEC-144	Mean
T1	100	100	100	100	100	100	100	100	100	100	100	100	100	100	100	100	100	100	100	100 ^a^ (±0.5)
T2	34.1	41.2	37.1	38.1	32.3	38.2	40.3	48.2	43.3	56.1	55.3	53.2	65.2	59.1	70.2	75.1	65.3	56.5	58.4	50.9 ^b^ (±3.7)
T3	30.0	28.2	26.4	27.3	28.1	32.1	22.3	29.1	22.3	27.1	42.1	37.2	41.1	35.1	65.1	74.2	49.2	48.1	41.2	37.8 ^c^ (±3.8)
T4	22.4	21.5	19.2	15.2	20.1	19.3	18.3	16	17.1	15.3	35.3	28.2	31.3	28.1	53.2	53.5	39.1	38.2	48.3	28.4 ^d^ (±3.4)
Mean (Temp.)	46.6 ^gh^ (±6.7)	47.7 ^g^ (±6.7)	45.7 ^h^ (±6.9)	45.1 ^h^ (±7.3)	45.1 ^h^ (±6.7)	47.4 ^g^ (±6.7)	45.2 ^h^ (±7.0)	48.3 ^f^ (±6.9)	45.7 ^h^ (±7.0)	49.6 ^f^ (±7.0)	58.2 ^de^ (±5.4)	54.7 ^e^ (±5.7)	59.4 ^d^ (±5.2)	55.6 ^e^ (±6.0)	72.1 ^b^ (±3.7)	75.7 ^a^ (±3.6)	63.4 ^c^ (±5.1)	60.7 ^d^ (±6.0)	62.0 ^cd^ (±5.0)	54.11
CD (*p* = 0.05); G = 1.04 **; T = 0.48 **; G × T = 2.09 **																				

where: T1 = 20 °C; T2 = 30 °C; T3 = 25–40 °C; T4 = 40 °C. Different letters on mean values depict significant differences (*p* ≤ 0.01), where cases of means are three (*n* = 3). The meaning of ** is significance at *p* ≤ 0.01.

**Table 2 plants-12-01400-t002:** Biochemical parameters at seedling stage in different groups of mustard cultivars.

Groups	Cultivars	Chlorophyll Content	Catalase Activity	Peroxidase Activity	Superoxide Dismutase	Proline Content
		(µg/g FW)	(µmol/g FW)	(µmol/g FW)	Activity(U/g FW)	(µmol/g FW)
Susceptible cultivars	PDZ-1	437 ^ef^ (±3.7)	21.9 ^d^ (±0.3)	338 ^d^ (± 4.8)	297 ^c^ (±2.8)	4.86 ^de^ (±0.05)
	PDZ-3	483 ^e^ (±3.0)	19.3 ^e^ (±0.1)	272 ^e^ (±1.9)	291 ^c^ (±1.9)	4.51 ^e^ (±0.05)
	PDZ-4	399 ^f^ (±5.2)	19.6 ^e^ (±0.2)	328 ^d^ (±2.8)	284 ^cd^ (±1.6)	4.45 ^f^ (±0.05)
	PDZ-5	395 ^f^ (±6.9)	17.3 ^f^ (±0.2)	298 ^de^ (±4.3)	276 ^d^ (±1.4)	4.55 ^e^ (±0.07)
	PDZ-7	496 ^e^ (±7.0)	16.0 ^f^ (±0.1)	310 ^d^ (±2.5)	285 ^cd^( ±1.8)	4.88 ^de^ (±0.06)
	PDZ-8	545 ^d^ (±2.9)	20.0 ^d^ (±0.1)	346 ^d^ (±3.5)	306 ^c^ (±1.3)	5.04 ^d^ (±0.05)
	PDZ-11	498 ^e^ (±5.3)	17.7 ^f^ (±0.2)	266 ^e^ (±2.0)	277 ^d^ (±1.9)	4.70 ^de^ (±0.04)
	PDZ-12	565 ^cd^ (±5.4)	22.9 ^d^ (±0.2)	333 ^d^ (±2.7)	311 ^bc^ (±1.8)	6.04 ^c^ (±0.05)
	Pusa Karishma	493 ^e^ (±4.4)	20.2 ^d^ (±0.1)	318 ^de^ (±3.8)	285 ^cd^ (±1.5)	4.52 ^e^ (±0.02)
	Heera	530 ^de^ (±6.7)	19.4 ^e^ (±0.2)	265 ^e^ (±2.7)	312 ^bc^ (±1.8)	4.68 ^de^ (±0.01)
Moderately tolerant cultivars	JC-21	487 ^e^ (±4.0)	29.7 ^c^ (±0.3)	469 ^c^ (±2.5)	333 ^b^ (±1.3)	5.06 ^c^ (±0.07)
	PM-21	589 ^c^ (±5.7)	28.5 ^c^ (±0.3)	418 ^cd^ (±4.5)	355 ^ab^ (±1.4)	6.89 ^c^ (±0.14)
	PM-22	659 ^b^ (±3.7)	26.7 ^cd^ (±0.4)	498 ^c^ (±4.8)	392 ^a^ (±2.5)	9.98 ^a^ (±0.17)
	Navgold	534 ^de^ (±4.6)	25.7 ^cd^ (± 0.1)	449 ^c^ (±4.4)	324 ^bc^ (±4.5)	6.79 ^c^ (±0.05)
Tolerant cultivars	PM-30	749 ^a^ (±5.9)	36.4 ^a^ (±0.3)	633 ^a^ (±5.4)	387 ^a^ (±1.4)	10.39 ^a^ (±0.10)
	Pusa Bold	739 ^a^ (±10.5)	37.3 ^a^ (±0.2)	633 ^a^ (±5.2)	404 ^a^ (±2.1)	10.12 ^a^ (± 0.09)
	Pusa Vijay	689 ^b^ (±5.9)	35.6 ^ab^ (± 0.1)	606 ^ab^ (±4.2)	360 ^b^ (±2.5)	9.14 ^ab^ (±0.16)
	JC-33	544 ^d^ (±5.8)	33.8 ^b^ (±0.4)	603 ^ab^ (±3.6)	374 ^ab^ (±1.9)	9.44 ^a^ (±0.08)
	BEC-144	585 ^c^ (±8.8)	35.1 ^ab^ (±0.6)	543 ^b^ (±1.6)	352 ^b^ (±2.1)	8.40 ^b^ (±0.07)
Grand Mean		548	25.4	417	327	6.55
CD (*p* =0.05)		7.97 **	0.33 **	4.08 **	4.25 **	0.11 **

Different letters superscripted on mean values depict significant differences (*p* ≤ 0.01) among the cultivars, where cases of means are three (*n* = 3). The meaning of ** is significance at *p* ≤ 0.01.

**Table 3 plants-12-01400-t003:** Correlation among the various physio-biochemical parameters measured in 19 mustard cultivars under the contrasting temperature conditions.

	Germination %	Survival Percentage	MGT	Vigor Index II	Chlorophyll Content	Catalase Activity	Peroxidase Activity	SOD Activity	Proline Content
Germination %	1.000								
Survival percentage	0.836 **	1.000							
MGT	−0.734 **	−0.575 **	1.000						
Vigor index II	0.837 **	0.756 **	−0.665 **	1.000					
Chlorophyll content	0.800 **	0.858 **	−0.636 **	0.816 **	1.000				
Catalase activity	−0.499 **	−0.402 **	0.598 **	−0.291 **	−0.430 **	1.000			
Peroxidase activity	−0.363 **	−0.226 **	0.448 **	−0.100 **	−0.198 **	0.905 **	1.000		
SOD activity	0.848 **	0.923 **	−0.666 **	0.832 **	0.925 **	−0.428 **	−0.212 **	1.000	
Proline content	−0.590 **	−0.544 **	0.647 **	−0.439 **	−0.559 **	0.909 **	0.824**	−0.576 **	1.000

** indicates a correlation significant at 1%.

**Table 4 plants-12-01400-t004:** Mean seed yield attributing traits of different groups of cultivars.

Cultivars	No. of Primary Branches	No. of Secondary Branches	Main Shoot Length (cm)	No. of Siliquae on Main Shoot	Siliqua Length (cm)	No. of Seeds per Siliqua	Siliqua Density on Main Shoot	Seed Yield per Plant (g)
Susceptible	8.5 ^c^ (±0.26)	22.8 ^c^ (±4.67)	50.6 ^c^ (±1.06)	31.6 ^c^ (±2.62)	3.37 ^c^ (±0.19)	13.7 ^c^ (±0.37)	0.74 ^c^ (±0.04)	31.9 ^c^ (±1.31)
Moderately tolerant	9.1 ^b^ (±0.09)	24.2 ^b^ (±1.20)	57.1 ^b^ (±2.22)	33.6 ^b^ (±1.13)	3.87 ^b^ (±0.26)	13.5 ^b^ (±1.65)	0.62 ^b^ (±0.08)	34.8 ^b^ (±1.66)
Tolerant	10.8 ^a^ (±0.26)	29.0 ^a^ (±0.88)	69.5 ^a^ (±2.83)	39.3 ^a^ (±0.75)	5.01 ^a^ (±0.11)	13.7 ^a^ (±1.61)	0.68 ^a^ (±0.06)	40.9 ^a^ (±2.39)
Mean	9.5	25.3	59.1	34.8	4.08	13.6	0.68	35.8
CD (*p* =0.05)	0.54 **	6.88 *	5.28 **	4.15 **	0.48 **	NS	NS	4.49 **

NS = non-significant; different letters superscripted on mean values depict significant differences (* and ** *p* < 0.05 and *p* < 0.01, respectively) among the cultivars, where cases of means are three (*n* = 3).

**Table 5 plants-12-01400-t005:** Mean weather data throughout the growing periods in *rabi* seasons of 2020–2021 and 2021–2022.

Months	Tmax (°C)	Tmin (°C)	Relative Humidity (%)	Rainfall (mm)	Sunshine (Hrs)
October	32.3	19.3	74.1	4.2	7.5
November	27.12	10.59	71.78	0	4.5
December	21.88	7.3	77.4	6	3.73
January	17.37	7.45	84	0	2.45
February	22.82	8.54	71.61	0	6.5
March	31.62	14.97	61.21	0	8.1

## Data Availability

Not applicable.

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
