# Peer review of "Heat-Stress-Induced Changes in Physio-Biochemical Parameters of Mustard Cultivars and Their Role in Heat Stress Tolerance at the Seedling Stage"

_plants, 2023, doi:10.3390/plants12061400_

Round 1

Reviewer 1 Report (Previous Reviewer 2)

Major points

1) The authors added the indication of statistical significance in the figures in the revised manuscript, but I regret to say the added letters in Figures 1 and 4 to 8 are not appropriate. The authors compared the four heat stress treatments and tested the significance of the differences among the treatments. But it is more important to examine the differences among the genotypes since this study focuses on the difference in heat stress tolerance among the genotypes. Therefore, I think the differences among the “susceptible”, “moderately tolerant”, and “tolerant” genotypes should be tested as in Table 4.

And in my opinion, it seems that there are no significant differences in germination percentage between the “tolerant” and “moderately tolerant” genotypes (Figure 4a). The differences in mean germination time also seems to be non-significant among the three genotype groups (Figure 4b). If so, the text should be revised accordingly.

Minor points

2) In the figure and table captions, the phrases “Same alphabets depict non-significant differences (p<0.05)” are not accurate, which should be changed to “Different alphabets depict significant difference (p<0.05).”

3) Asterisks should be deleted from Table 3 since they don’t indicate statistical significance. The width of the column “MGT” is not wide enough and should be extended,

Author Response

Major points

Comment 1:  The authors added the indication of statistical significance in the figures in the revised manuscript, but I regret to say the added letters in Figures 1 and 4 to 8 are not appropriate. The authors compared the four heat stress treatments and tested the significance of the differences among the treatments. But it is more important to examine the differences among the genotypes since this study focuses on the difference in heat stress tolerance among the genotypes. Therefore, I think the differences among the “susceptible”, “moderately tolerant”, and “tolerant” genotypes should be tested as in Table 4. And in my opinion, it seems that there are no significant differences in germination percentage between the “tolerant” and “moderately tolerant” genotypes (Figure 4a). The differences in mean germination time also seems to be non-significant among the three genotype groups (Figure 4b). If so, the text should be revised accordingly.

Reply 1: Thank you for the critical inputs, since the paper discusses the effect of treatment, genotype and their interaction, the added letters in Figure 1 and 4 to 8 are indicating the significant differences among the treatments within a group and not among the treatments and thus are relevant. Moreover, the differences among physiological parameters in the “susceptible”, “moderately tolerant”, and “tolerant” genotypes has been already discussed in the results and given in Figure 3. The observation regarding germination is correct and we have made the necessary revisions with respect to MGT in the manuscript.

Minor points

Comment 2:  In the figure and table captions, the phrases “Same alphabets depict non-significant differences (p<0.05)” are not accurate, which should be changed to “Different alphabets depict significant difference (p<0.05).”

Reply: The statement "Similar alphabets represent non-significant differences" has been removed from the figure and table captions as the reviewer advised.

Comment 3: Asterisks should be deleted from Table 3 since they don’t indicate statistical significance. The width of the column “MGT” is not wide enough and should be extended,

Reply: The Asterisks inTable 3 indicate that the value is significant at 1% level. The width of the column “MGT” has been extended as per the suggestions.

The reviewer and editors comments are reasonable, and we have corrected the MS in accordance with the comments and suggestions. A thorough internal reviews was performed in the whole MS, changes highlighted in Track Change Format supplied MS. We are thankful to learned reviewer for giving critical insights, leading to substantial improvement in the manuscript, we hope the response meets the reviewer and editor approval.

Reviewer 2 Report (New Reviewer)

The manuscript deals with the physiological and biochemical responses of mustard to heat stress. The paper requires substantial English revision. In the current form it is hard to follow and many statements are meaningless. Some words are not suitable to scientific paper. Check the values from the text with Tables and Figures in the Results because some of them are different. Only some comments are listed below:

L3: use the term cultivar or variety instead genotype, throughout the paper

L3: ‘and their role in its tolerance’ – rephrase

L22-24: indicate % changes of examined parameters

L26: ‘based grouping confirmed it’ – rephrase

L26-27: what three single zero and double zero mean?

L30: antioxidant system activity

L32: ‘than the remaining single and double zero’ – how many?

L34: ‘Identification of heat stress tolerant donor’s based’ – replace with ‘Identification of heat stress tolerant cultivars’

L35: ‘be done and be included’ – replace with be created and included

L36: replace ‘technique’ with cultivars

L41-42: rephrase

L44: remove systems

L71: describe briefly these systems. Include antioxidant enzymes (CAT, POD, SOD) and non-enzymatic antioxidants (glutathione and ascorbate) which are ubiquitous in different abiotic (heat, drought, pesticides, heavy metals) and biotic (pathogens) stress mitigation. The Authors can include the following references: https://doi.org/10.1016/j.scienta.2022.110988 and https://doi.org/10.1007/s00425-022-03838-x

L72: Mustard is one of the most important edible oilseed crop worldwide

L75: ‘occupied’ – rephrase

L78: ‘establishment’ – rephrase

L81: replace ‘repress’ with ‘inhibit’

L110-111: indicate exact values with the Table 1. First description of the Results refers to the Table 1, therefore, it should be placed before Fig. 1

L141: replace ‘alphabets’ with letters, ‘SEm’ with ‘SEM’ and ‘non-significant’ with no significant. Correct in all Tables and Figures

L142: listed cultivars should be grouped in susceptible, moderately tolerant and tolerant

L167: replace ‘(d)’ with ‘(days)’

L194-196: clarify this statement and make it corresponding with points on the Fig. 5

L311: Physiological parameters of seedlings

L326-327: remove ‘n’ and replace ’0<r ≤1’ with exact p value

L399, 408: remove seed

L403, 435: replace ‘seeds’ with ‘seedlings’

L422-423: use the terms susceptible, moderately tolerant and tolerant instead ‘double zero genotype’

L441: what the Authors mean by ‘conventional and quality mustard’?

L444: remove ‘viz.’ throughout the paper and replace with : or following

L445-446: indicate what is the difference between these cultivars

L455: replace ‘Walk-in germinator’ with ‘phytotron room’

L455-456 and 460: remove ‘in the Division of Seed Science 455 and Technology, ICAR-IARI, New Delhi (India)’ and ‘of the National Phytotron Facility (NPF), ICAR-IARI, New Delhi (India)’ – it is not necessary for experiment under controlled conditions.

L476: Indicate what volume of water was used for watering and how often the seedlings were watered

L476: indicate full name for ‘FYM’

L484-485: indicate a full name for ‘NPF’. Precise that it is only the T3 treatment. Include also other treatments in the Figure. Replace time periods (e.g. 4-8 pm) with number of hours (e.g. 4 h) on the scheme

L487-488: rephrase. What the Authors mean by normal and abnormal seedlings?

L494: indicate the name of the Author before [27]

L500: abbreviate fresh weight as FW and remove the abbreviation from L501

L511: remove ‘fresh and healthy’

L519: remove ‘fresh’ and indicate the range of seedlings weight if the Authors express the results per FW

L547: remove ‘fresh’ and indicate seedling weight

L557: replace ‘Seed yield attributing traits’ with ‘Physiological parameters of seedlings’. L558-583: Combine all these parameters in one point: ‘Physiological parameters of seedlings’. Description of each parameter begin in a new paragraph

L562, 567, 571, etc. – what the Authors mean by siliqua?

L585: remove ‘field’ – the experiment was performed under controlled conditions

L592: remove ‘for Windows’ and indicate the city and country of the software manufacturer

L595: ‘appropriate software’ – indicate the software or omit if it is SPSS 16.0

L595: ‘dissimilarity matrix’ – rephrase

L602-604: summarize in general using the terms susceptible, moderately tolerant and tolerant instead ‘single zero genotype’ or ‘double zero genotype’

Author Response

To Editor/Reviewer #2

The manuscript deals with the physiological and biochemical responses of mustard to heat stress. The paper requires substantial English revision. In the current form it is hard to follow and many statements are meaningless. Some words are not suitable to scientific paper. Check the values from the text with Tables and Figures in the Results because some of them are different. Only some comments are listed below:

Comment :  L3: use the term cultivar or variety instead genotype, throughout the paper.

Reply: The word "cultivar" has been used in place of "genotype" throughout the body of the manuscript.

Comment :  L3: ‘and their role in its tolerance’ – rephrase

Reply: It has been modified accordingly.

Comment :  L26: ‘based grouping confirmed it’ – rephrase

Reply: It has been modified accordingly.

Comment :  L30: antioxidant system activity.

Reply: The word has been modified accordingly.

Comment :  L32: ‘than the remaining single and double zero’ – how many?

Reply: The numbers have been added to the sentence.

Comment :  L34: ‘Identification of heat stress tolerant donor’s based’ – replace with ‘Identification of heat stress tolerant cultivars’

Reply: Thank you for your suggestion. We have improved the sentence in response to your recommendations, please.

Comment :  L35: ‘be done and be included’ – replace with be created and included.

Reply: The words have been replaced with selected and included.

Comment :  L36: replace ‘technique’ with cultivars.

Reply: The word has been replaced according to the suggestions.

Comment :  L41-42: rephrase.

Reply: The sentence has been rephrased accordingly.

Comment :  L44: remove systems.

Reply: It has been removed accordingly.

Comment :  L71: describe briefly these systems. Include antioxidant enzymes (CAT, POD, SOD) and non-enzymatic antioxidants (glutathione and ascorbate) which are ubiquitous in different abiotic (heat, drought, pesticides, heavy metals) and biotic (pathogens) stress mitigation. The Authors can include the following references: https://doi.org/10.1016/j.scienta.2022.110988 and https://doi.org/10.1007/s00425-022-03838-x

Reply: Thank you for your suggestion. The sentence has been changed in accordance with the comments and the sources provided.

Comment :  L72: Mustard is one of the most important edible oilseed crop worldwide.

Reply: The sentence has been rephrased accordingly.

Comment :  L75: ‘occupied’ – rephrase.

Reply: The sentence has been rephrased accordingly.

Comment :  L78: ‘establishment’ – rephrase.

Reply: The sentence has been rephrased accordingly.

Comment :  L81: replace ‘repress’ with ‘inhibit’.

Reply: The word has been replaced accordingly.

Comment :  L110-111: indicate exact values with the Table 1. First description of the Results refers to the Table 1, therefore, it should be placed before Fig. 1.

Reply: The results description has been corrected as per suggestions.

Comment :  L141: replace ‘alphabets’ with letters, ‘SEm’ with ‘SEM’ and ‘non-significant’ with no significant. Correct in all Tables and Figures.

Reply: All tables and figures now have been improved and replaced appropriately in the text.

Comment :  L142: listed cultivars should be grouped in susceptible, moderately tolerant and tolerant.

Reply: The cultivars have been grouped accordingly.

Comment :  L167: replace ‘(d)’ with ‘(days)’.

Reply: The word has been replaced accordingly.

Comment :  L194-196: clarify this statement and make it corresponding with points on the Fig. 5.

Reply:  The changes have been done as per the suggestions.

Comment :  L311: Physiological parameters of seedlings.

Reply: These are yield attributing traits and thus, it will be incorrect to call them Physiological parameters of seedlings.

Comment :  L326-327: remove ‘n’ and replace ’0<r ≤1’ with exact p value.

Reply: It has been replaced accordingly.

Comment :  L399, 408: remove seed.

Reply: It has been removed accordingly.

Comment :  L403, 435: replace ‘seeds’ with ‘seedlings’.

Reply: It has been replaced accordingly.

Comment :  L422-423: use the terms susceptible, moderately tolerant and tolerant instead ‘double zero genotype’

Reply: The sentence has been modified accordingly.

Comment :  L441: what the Authors mean by ‘conventional and quality mustard’?

Reply:  The conventional and quality mustard has been explained in the introduction section and also in the materials and methods.

Comment :  L444: remove ‘viz.’ throughout the paper and replace with: or following.

Reply: It  has been removed accordingly.

Comment :  L445-446: indicate what is the difference between these cultivars.

Reply: The difference has been explained.

Comment :  L455: replace ‘Walk-in germinator’ with ‘phytotron room’

Reply: Walk-in germinator is a technical term routinely used in such experiments.

Comment :  L455-456 and 460: remove ‘in the Division of Seed Science  and Technology, ICAR-IARI, New Delhi (India)’ and ‘of the National Phytotron Facility (NPF), ICAR-IARI, New Delhi (India)’ – it is not necessary for experiment under controlled conditions.

Reply: ‘in the Division of Seed Science  and Technology’ has been removed accordingly.

Comment :  L476: Indicate what volume of water was used for watering and how often the seedlings were watered.

Reply: Necessary details have been included accordingly.

Comment :  L476: indicate full name for ‘FYM’.

Reply: In the manuscript, full name for FYM is given.

Comment :  L484-485: indicate a full name for ‘NPF’. Precise that it is only the T3 treatment. Include also other treatments in the Figure. Replace time periods (e.g. 4-8 pm) with number of hours (e.g. 4 h) on the scheme.

Reply: The corrections have been made in the manuscript as per suggestions.

Comment :  L487-488: rephrase. What the Authors mean by normal and abnormal seedlings?

Reply: International Seed Testing Association (ISTA) is an organization which works for harmonization of seed testing methods at international level. According to ISTA, the germination test results of mustard are categorized into normal seedlings (germination%), abnormal seedlings, dead and hard seeds.  The definition of normal and abnormal seedlings is from the ISTA rules, 2020 is reproduced below for your kind perusal.

Normal Seedling: Normal seedlings show the potential for continued de­velopment into satisfactory plants when grown in good quality soil and under favourable conditions of moisture, temperature and light (ISTA Rule no, 5.2.7).

Abnormal Seedling: Abnormal seedlings do not show the potential to develop into a normal plant when grown in good quality soil and under favourable conditions of moisture, temperature and light (ISTA, Rule no, 5.2.8).   

Comment :  L494: indicate the name of the Author before [27]

Reply: It has been added.

Comment :  L500: abbreviate fresh weight as FW and remove the abbreviation from L501.

Reply: The corrections have been made in the manuscript as per suggestions in seedling DW measurements.

Comment :  L511: remove ‘fresh and healthy’.

Reply: It  has been removed accordingly.

Comment :  L519: remove ‘fresh’ and indicate the range of seedlings weight if the Authors express the results per FW.

Reply: It  has been removed accordingly.

Comment :  L547: remove ‘fresh’ and indicate seedling weight.

Reply: Corrections has been done accordingly.

Comment :  L557: replace ‘Seed yield attributing traits’ with ‘Physiological parameters of seedlings’. L558-583: Combine all these parameters in one point: ‘Physiological parameters of seedlings’. The description of each parameter begin in a new paragraph.

Reply: The seed yield attributing traits and Physiological parameters of seedlings are two different aspects Therefore, combining all these parameters would give the wrong interpretation, please.  

Comment :  L562, 567, 571, etc. – what the Authors mean by siliqua?

Reply: Siliqua or siliquae is the internationally accepted term/name of cruciferous fruit/pod.

Comment :  L585: remove ‘field’ – the experiment was performed under controlled conditions.

Reply: It is kindly informed that the experiment was conducted in controlled condition as well as in the field. The field data has been added in the response of reviewer 1’s suggestion, thus, the word "field" has also been added to the experimental details.

Comment :  L592: remove ‘for Windows’ and indicate the city and country of the software manufacturer

Reply: Required information has been provided in the manuscript.

Comment :  L595: ‘appropriate software’ – indicate the software or omit if it is SPSS 16.0

Reply: The corrections have been done as per the suggestions.

Comment :  L595: ‘dissimilarity matrix’ – rephrase.

Reply: UPGMA is a clustering technique that uses the (unweighted) arithmetic averages of the measures of dissimilarity, thus avoiding characterizing the dissimilarity by extreme values (minimum and maximum) between the considered genotypes and thus, the dissimilarity matrix was used.

Comment :  L602-604: summarize in general using the terms susceptible, moderately tolerant and tolerant instead ‘single zero genotype’ or ‘double zero genotype’

Reply: It has been summarized accordingly.

The reviewer and editors comments are reasonable, and we have corrected the MS in accordance with the comments and suggestions. A thorough internal reviews was performed in the whole MS, changes highlighted in Track Change Format supplied MS. We are thankful to learned reviewer for giving critical insights, leading to substantial improvement in the manuscript, we hope the response meets the reviewer and editor approval.

Round 2

Reviewer 1 Report (Previous Reviewer 2)

The authors’ have not modified the figures in the revised version. I’m not fully convinced by the authors’ rebuttal and I still think that it would be better to compare differences among the three cultivar groups in Figures 1 and 4 to 8.

However, the significant differences among individual cultivars in survival percentage and the biochemical parameters are already shown in Tables 1 and 2. Therefore, I’m not further insisting on my opinion.

Author Response

Author responses to editor/reviewer (1) comments

(plants-2235694)

To Editor/Reviewer #1

Comment: The authors’ have not modified the figures in the revised version. I’m not fully convinced by the authors’ rebuttal and I still think that it would be better to compare differences among the three cultivar groups in Figures 1 and 4 to 8.

Reply: We have mentioned all the corrections inside the figures as suggested.

Comment: However, the significant differences among individual cultivars in survival percentage and the biochemical parameters are already shown in Tables 1 and 2. Therefore, I’m not further insisting on my opinion.

Reply: Thank you for your positive comments.

The reviewer and editors comments are reasonable, and we have corrected the MS in accordance with the comments and suggestions. A thorough internal reviews was performed in the whole MS, changes highlighted in Track Change Format supplied MS. We are thankful to learned reviewer for giving critical insights, leading to substantial improvement in the manuscript, we hope the response meets the reviewer and editor approval.

Reviewer 2 Report (New Reviewer)

The Authors have corrected the paper according to the comments. However, it still needs English revision. I have also other comments listed below:

L156: 'where cases of means are three: n = 3' - replace by (n=3) in all Figures and Tables

L251: in Table 2 indicate which cultivars are susceptible, moderately tolerant and tolerant

L408: replace seed with seedling

L409: 'including POD, SOD and CAT to lessen the harm caused by ROS' include also non-enzymatic antioxidants (https://doi.org/10.1016/j.scienta.2022.110988)

L510: 'National Phytotron Facility' - remove

L540: 'of National Phytotron Facility (NPF)' - remove

L543: 'A seedling was considered to be normal if its shoot and root growth were normal' - rephrase

L616: 'Seed yield attributing traits' - replace with 'Yield parameters'

L645: the presented studies do not imply that some of them were conducted in field conditions. Authors should indicate which part concerns field studies and present weather conditions

L661: 'The results of this present study stipulated' - rephrase

Author Response

Author responses to editor/reviewer (2) comments

(plants-2235694)

To Editor/Reviewer #2

Comment: L156: 'where cases of means are three: n = 3' - replace by (n=3) in all Figures and Tables.

Reply: We have corrected all the suggestion in the text.

Comment: L251: in Table 2 indicate which cultivars are susceptible, moderately tolerant and tolerant

Reply: It has been mentioned

Comment: L408: replace seed with seedling

Reply: We have replaced in the text.

Comment: L409: 'including POD, SOD and CAT to lessen the harm caused by ROS' include also non-enzymatic antioxidants (https://doi.org/10.1016/j.scienta.2022.110988)

Reply: We have corrected all the suggestion in the text as well as references.

Comment: L510: 'National Phytotron Facility' - remove

Reply: We have corrected all the suggestion in the text.

Comment: L540: 'of National Phytotron Facility (NPF)' - remove

Reply: We have corrected all the suggestion in the text.

Comment: L543: 'A seedling was considered to be normal if its shoot and root growth were normal' - rephrase

Reply: We have rephrased and the corrections have been inside the text.

Comment: L616: 'Seed yield attributing traits' - replace with 'Yield parameters'

Reply: We have corrected all the suggestion in the text.

Comment: L645: the presented studies do not imply that some of them were conducted in field conditions. Authors should indicate which part concerns field studies and present weather conditions

Reply: According to the reviewers' comments, additional yield parameters for different groups of cultivars were included in the current study. All necessary field and weather-related information have now been provided.

Comment: L661: 'The results of this present study stipulated' – rephrase

Reply: We have corrected all the suggestion in the text.

The reviewer and editors comments are reasonable, and we have corrected the MS in accordance with the comments and suggestions. A thorough internal reviews was performed in the whole MS, changes highlighted in Track Change Format supplied MS. We are thankful to learned reviewer for giving critical insights, leading to substantial improvement in the manuscript, we hope the response meets the reviewer and editor approval.

Round 3

Reviewer 2 Report (New Reviewer)

The Authors have improved the paper according to the comments. I have no more questions.

This manuscript is a resubmission of an earlier submission. The following is a list of the peer review reports and author responses from that submission.

Round 1

Reviewer 1 Report

The manuscript titled "Thermopriming induced changes in physio-biochemical parameters and their role in heat stress tolerance mustard genotypes at the seedling stage" by Sakpal et al. studied some physiological and biochemical traits of Indian mustard genotypes that contribute to heat stress adaptation; this was performed at germination and early seedling stages. The work provides a good information as ranking of 19 Indian mustard genotypes to heat stress is concerned based on germination and early seedling growth. As our ultimate goal from the research carried out on the crops is their productivity, it would be better if yield parameter was included in the ranking of these genotypes. The below issues need to be addressed to improve the submission before publication.

-          In the title, thermo priming should be one word (i.e., thermopriming).

-          Two important growth conditions should be provided for reproducibility purposes: the photoperiod and photosynthetic photon flux density (PPFD) in µmol m-2 s-1.

-          For the spectrophotometer used, model, manufacturer name and country should be provided.

-          The quality of the figures needs to be improved. The Y-axis of Figs. 1, 3, 4, and 5 are not labelled. Figs. 5 and 6 should be combined in one figure (i.e., Fig. 5). The same for figs. 7 and 8 (i.e., Fig. 6).

-          At the end of each measured variable presented in the result’s section, the authors added brief explanation of the results. This is not accepted as no interpretation of the data in the result’s section. Please leave out.

-          The cited literature throughout the manuscript is not at the cutting edge of the science. Out of 67 citations, only 7 are from the last five years. Therefore, the citations included in the manuscript should be extensively updated relative to relevant recent literature.

Reviewer 2 Report

This study examined the effect of heat stress on seedling survival rate in 19 mustard genotypes. The authors classified the 19 genotypes into three groups regarding heat tolerance, and also examined the changes in germination and seedling growth, chlorophyll and proline contents, and activities of antioxidant defense enzymes. They succeeded to screen tolerant genotypes by a simple method based on survival rate, but this is not a novel method at all. The changes in biochemical parameters presented in this manuscript are quite typical stress responses, which are well-known and not novel findings. I understand that this study is an important first step for breeding of heat tolerant mustard. But this study does not provide new insight into the mechanism of heat tolerance, and I regret to say the physiological significance of this study is low.

In addition, I don’t understand that the data presented in this manuscript are significant since there are no indication of statistical significance in the figures. Therefore, I cannot evaluate whether the authors’ claims are experimentally supported or not.

I also point out several concerns about this manuscript as follows.

1) The phrase “Thermo priming” in the title is incorrect. This study performed only selection of heat tolerant genotypes and their characterization, which is not related to priming at all.

2) As I mentioned above, there are no indication of statistical significance in Figures, which should be presented. In Table1, the statistical significance between the individual genotypes should be presented so that you can indicate that the genotypes with high survival rate are truly heat tolerant.

3) In Abstract, the authors mention that several single zero and double zero genotypes (PM-21, PM- 22, PM-33, JC-21 and JC-33) exhibited more efficient antioxidant system and proline accumulation than the remaining single and double zero genotypes. Since there are no data of these parameters of individual genotypes, the authors’ claim is not very convincing.

4) The authors classified the 19 genotypes into three groups by survival rate shown in Table 1. And they also categorized the samples into three groups by clustering based on biochemical parameters in Figure 2. I don’t understand the relation of these groups. The authors mention that Group III corresponds to the tolerant genotypes, but is it true? Do the members of tolerant genotypes also belong to Group III?

Moreover, as the authors selected heat tolerant genotypes by survival rate, I don’t understand the reason why they also categorized the samples based on the biochemical parameters. I think selection of tolerant genotypes based on the data of biochemical analyses is not an efficient and practical way.

5) Although the authors calculated correlation among various parameters, Figure 9 does not indicate them inadequately. The correlation should be presented as a heat map or as a table so that the values of correlation are visible.